# Adaptive Auxiliary Task Weighting for Reinforcement Learning

Xingyu Lin*        Harjatin Singh Baweja*        George Kantor        David Held

Robotics Institute
Carnegie Mellon University
{xlin3, harjatis, kantor, dheld}@andrew.cmu.edu

## Abstract

Reinforcement learning is known to be sample inefficient, preventing its application to many real-world problems, especially with high dimensional observations like images. Transferring knowledge from other auxiliary tasks is a powerful tool for improving the learning efficiency. However, the usage of auxiliary tasks has been limited so far due to the difficulty in selecting and combining different auxiliary tasks. In this work, we propose a principled online learning algorithm that dynamically combines different auxiliary tasks to speed up training for reinforcement learning. Our method is based on the idea that auxiliary tasks should provide gradient directions that, in the long term, help to decrease the loss of the main task. We show in various environments that our algorithm can effectively combine a variety of different auxiliary tasks and achieves significant speedup compared to previous heuristic approaches of adapting auxiliary task weights.

## 1   Introduction

Deep reinforcement learning has enjoyed recent success in domains like games [1, 2], robotic manipulation, and locomotion tasks [3, 4]. However, most of these applications are either limited to simulation or require a large number of samples collected from real-world experiences. For complex tasks, the reinforcement learning algorithm often requires a prohibitively large number of samples to learn the policy [5, 6]. The sample complexity is even worse when learning from image observations, in which more samples are needed to learn a good feature representation.

Transferring knowledge from other tasks can be a powerful tool for learning efficiently. Two types of transferring are often used: representational and functional transfer [7]. In representational transfer, a representation previously learned from other tasks is used for the task at hand. For example, visual features can be taken from pre-trained features of other tasks, such as image classification or depth estimation [8]. However, visual features learned from these static tasks might not be useful features for decision-making tasks studied in reinforcement learning. Further, the visual environment encountered by the reinforcement learning agent might have a different appearance from which these visual features were trained, thus limiting the benefit of such transfer.

In functional transfer, multiple tasks sharing the representation are trained jointly. In the context of reinforcement learning, auxiliary tasks are trained jointly with the reinforcement learning task [9, 10, 11], as illustrated in Figure 1. This approach has the advantage that the learned representation will be relevant to the environment in which the agent is operating. Many self-supervised tasks proposed in previous works can be used [9, 11, 12, 13, 14].

The challenge for using auxiliary tasks is to select what set of auxiliary tasks to use and to determine the weighting of different auxiliary tasks; some auxiliary tasks may be more or less relevant to the

---

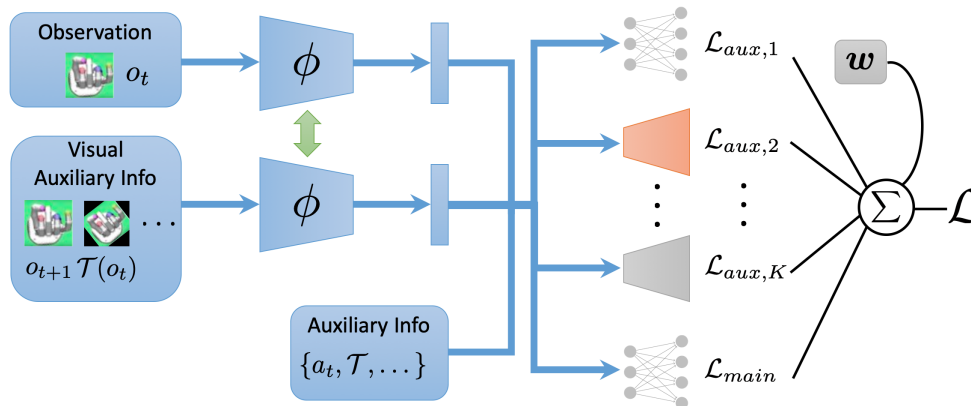

Figure 1: An illustration of learning with auxiliary tasks. All of the visual observations and any auxiliary visual information are passed through a shared weight CNN to get a low dimension representation. The representation, along with other auxiliary information are then used to perform different auxiliary tasks, as well as the main task. A final loss is computed by weighting the main loss and all the auxiliary losses.

reinforcement learning objective. Further, the usefulness of an auxiliary task may change over the course of the learning process: one auxiliary task might be useful for learning a feature representation for reinforcement learning in the beginning of training, but it might no longer be useful later in training. Some auxiliary tasks may even slow down reinforcement learning if weighted too strongly. While auxiliary tasks have been commonly used for learning a feature representation, it is still an open question to determine which auxiliary tasks to use [15]. Previous work either relies on prior knowledge about the tasks [8, 16] or uses a grid search to determine the best weighting parameters, which requires repeatedly learning the same task with different hyper-parameters.

We meet this challenge from a different perspective: instead of pre-determining the importance of each auxiliary task before the training begins, we can adjust the weight of each task online during the training process. In this way, we exploit the information obtained after actually applying the auxiliary tasks while performing reinforcement learning. Our framework follows the principle that a useful auxiliary task should provide a gradient that points in a good direction such that, in the long term, the loss of the reinforcement learning objective decreases.

In this work, we apply the above principle and propose a simple online learning algorithm that dynamically determines the importance of different auxiliary tasks for the specific task at hand. We show in various environments that our algorithm can effectively combine a variety of different auxiliary tasks and achieves significant speedup compared to previous heuristic approches of adapting auxiliary task weights.

## 2 Related Work

### 2.1 Auxiliary Tasks for Reinforcement Learning

While reinforcement learning with low-dimensional state input can also benefit from auxiliary tasks in partially observable environments [17, 18], auxiliary tasks have been more commonly used for reinforcement learning from images or other high-dimensional sensor readings. The auxiliary tasks can either be supervised learning tasks such as depth prediction [19], reward prediction [20], or task specific prediction [10]. Here we focus on self-supervised tasks where labels can be acquired in a self-supervised manner. Other works use alternative control tasks as auxiliary tasks, such as maximizing pixel changes or network features [9]. The weights for each of the auxiliary tasks are either manually set using prior intuition or through hyper-parameter tuning by running the full training procedure multiple times. In this work, we propose to determine what auxiliary tasks are useful at each time step of the training procedure and adaptively tune the weights, eliminating the hyper-parameter tuning, which becomes much harder as the number of auxiliary tasks grows.

While we use auxiliary tasks for learning a feature representation, auxiliary tasks can be used in other ways as well. For example, some works use auxiliary tasks for better exploration [21, 22]. These types of methods are orthogonal to the use of the auxiliary tasks in our work.

## 2.2 Adaptive Weights for Multiple Losses

The most similar analogy of auxiliary tasks in the context of supervised learning is multi-task learning [7, 15, 23]. Past work has found that, by sharing a representation among related tasks and jointly learning all the tasks, better generalization can be achieved over independently learning each task [24]. While MTL focuses on simultaneously learning multiple tasks, in this work, we only have one main task that we care about and the auxiliary tasks are only used to help learn the main task.

Some works in MTL assumes the amount of knowledge to transfer among tasks, i.e. task relationships to be known as a prior [25], or learned from the relationships among task specific parameters [26, 27]. In comparison, our work aims to adapt the weights of the auxiliary tasks online, to adaptively learn how much should we transfer from one auxiliary task to the main task, by looking at the gradient different tasks have on the shared parameters. As such, for future works, our method can also be applied to MTL problems after adapting our method from caring about only the transfer from auxiliary tasks to the main task, to caring about the transfer from one task to all the other tasks.

Other works involving multiple losses assume that all auxiliary tasks matter equally and adapt the weights based on the gradient norm [28] or task uncertainty [29]. A similar approach for anytime prediction balances the weights based on the average loss over the previous training time [30]. These methods assume that all of the auxiliary tasks are equally important. However, if we scale the number of auxiliary tasks, it is highly probable that some tasks will be more useful than others. In contrast, our method evaluates the usefulness of each task online and adapts the weights accordingly such that the more useful tasks receive a higher weight.

## 2.3 Learning Meta-parameters with Gradient Descent

An early work first proposed the idea of using online cross-validation with gradient descent to learn meta-parameters, which was also referred to as "adaptive bias" [31]. Some recent works can be viewed as instantiations of this method, where the meta-parameters are the parameters of the return function [32] or an intrinsic reward function [33]. Our method can be viewed as another variant of online cross-validation, where we treat the auxiliary task weights as meta-parameters and optimize over the weights for online representation learning.

# 3 Problem Definition

Assume that we have a main task $\mathcal{T}_{main}$ that we want to complete and a set of auxiliary tasks $\mathcal{T}_{aux,i}$, where $i \in \{1, 2, ..., K\}$. Each task has a corresponding loss $\mathcal{L}_{main}$ and $\mathcal{L}_{aux,i}$. In the context of reinforcement learning, $\mathcal{L}_{main}$ can either be the the expected return loss $\mathcal{L}_\pi$ which is used for calculating policy gradient, or the Bellman error $\mathcal{L}_Q$, such as for Q-learning. The losses are functions of all the model parameters $\theta_t$ at each training time step $t$.

Our goal is to optimize the main loss $\mathcal{L}_{main}$. However, using gradient-based optimization with only the main task gradient $\nabla_\theta \mathcal{L}_{main}$ is often slow and unstable, due to the high variance of reinforcement learning. Thus, auxiliary tasks are commonly used, especially for image based tasks, to help to learn a good feature representation. We can combine the main loss with the loss from the auxiliary tasks as

$$\mathcal{L}(\theta_t) = \mathcal{L}_{main}(\theta_t) + \sum_{i=1}^{K} w_i \mathcal{L}_{aux,i}(\theta_t), \tag{1}$$

where $w_i$ is the weight for auxiliary task $i$ and $\theta_t$ is the set of all model parameters at training step $t$. We assume that we update the parameters $\theta_t$ using gradient descent on this combined objective:

$$\theta_{t+1} = \theta_t - \alpha \nabla_{\theta_t} \mathcal{L}(\theta_t). \tag{2}$$

If a large number of auxiliary tasks are used, some auxiliary tasks may be more beneficial than others for learning a feature representation for the main task; thus the weights $w_i$ of each auxiliary

task (Eqn. 1) need to be tuned. Previous work manually tunes the auxiliary task weights $w_i$ [9]. However, a number of issues arise when we try to scale the number of auxiliary tasks. First, tuning the parameters $w_i$ becomes more computationally intensive as the number of auxiliary tasks $K$ increases. Second, if the values of $w_i$ are learned via hyperparameter optimization, then the reinforcement learning optimization must be run to near-convergence multiple times to determine the optimal values of $w_i$; ideally the weights would be learned online so that the reinforcement learning optimization only needs to be performed once. Last, the importance of each auxiliary task, and hence the optimal weight $w_i$, might change throughout the learning process; using a fixed value for $w_i$ might limit the performance.

# 4 Approach

We propose to dynamically adapt the weights for each auxiliary task. Below we describe our approach to doing so. We will first describe an approach that uses a one-step gradient in Section 4.1; we will then extend this framework to N-step gradient in Section 4.2.

## 4.1 Local Update from One-step Gradient

In our initial approach, we aim to find the weights for the auxiliary tasks such that $\mathcal{L}_{main}$ decreases the fastest. Specifically, define $\mathcal{V}_t(\boldsymbol{w})$ as the speed at which the main task loss decreases at the time step $t$, where $\boldsymbol{w} = [w_1, ..., w_k]^T$. We then have

$$
\begin{aligned}
\mathcal{V}_t(\boldsymbol{w}) = \frac{d\mathcal{L}_{main}(\theta_t)}{dt} &\approx \mathcal{L}_{main}(\theta_{t+1}) - \mathcal{L}_{main}(\theta_t) \\
&= \mathcal{L}_{main}(\theta_t - \alpha \nabla_{\theta_t} \mathcal{L}(\theta_t)) - \mathcal{L}_{main}(\theta_t) \\
&\approx \mathcal{L}_{main}(\theta_t) - \alpha \nabla_{\theta_t} \mathcal{L}_{main}(\theta_t)^T \nabla_{\theta_t} \mathcal{L}(\theta_t) - \mathcal{L}_{main}(\theta_t) \\
&= -\alpha \nabla_{\theta_t} \mathcal{L}_{main}(\theta_t)^T \nabla_{\theta_t} \mathcal{L}(\theta_t),
\end{aligned}
\tag{3}
$$

where $\alpha$ is the gradient step size. The first line is obtained from a finite difference approximation of the time derivative, with $\Delta t = 1$ (where $t$ is the iteration number of the learning process). The third line is a first-order Taylor approximation.

To update $\boldsymbol{w}$, we can simply calculate its gradient:

$$
\frac{\partial \mathcal{V}_t(w_i)}{\partial w_i} = -\alpha \nabla_{\theta_t} \mathcal{L}_{main}(\theta_t)^T \nabla_{\theta_t} \mathcal{L}_{aux,i}(\theta_t), \forall i = 1, ..., K.
\tag{4}
$$

This leads to an update rule that is based on the dot product between the gradient of the auxiliary task and the gradient of the main task. Intuitively, our approach leverages the online experiences to determine if an auxiliary task has been useful in decreasing the main task loss. The form of this equation resembles recent work which uses a thresholded cosine similarity to determine whether to use each auxiliary task [34]; however, our update rule is a product of our derivation of maximizing the speed at which the main task loss decreases and we will later show that it outperforms this method.

## 4.2 N-step Update

The gradient in Eqn. 4 is optimized for the instantaneous update rate of the main task, $d\mathcal{L}_{main}(\theta_t)/dt$, as shown in Equation 3. However, we are not actually concerned with the one-step update of the main task loss $\mathcal{L}_{main}(\theta_t)$; rather, we are concerned with the long-term value of $\mathcal{L}_{main}(\theta_t)$ after multiple gradient updates. In this section, we extend the method of the previous section to obtain an optimization objective for $\boldsymbol{w}$ that accounts for the performance on $\mathcal{L}_{main}(\theta_t)$ in the longer term.

Since the loss changes in one step does not necessarily reflect the long-term performance, we instead seek to optimize the N-step decrease of the main task loss:

$$
\mathcal{V}_t^N(\boldsymbol{w}) = \mathcal{L}_{main}(\theta_{t+N}) - \mathcal{L}_{main}(\theta_t).
$$

Exact computation of the gradient with respect to $\boldsymbol{w}$ requires calculating higher order Jacobians, which can be computationally expensive. We thus adopt a first-order approximation:

$$\mathcal{V}_t^N(\boldsymbol{w}) \doteq \mathcal{L}_{main}(\theta_{t+N}) - \mathcal{L}_{main}(\theta_t) \tag{5}$$

$$= \mathcal{L}_{main}\Big(\theta_{t+N-1} - \alpha\nabla_{\theta_{t+N-1}}\mathcal{L}(\theta_{t+N-1})\Big) - \mathcal{L}_{main}(\theta_t)$$

$$\approx \mathcal{L}_{main}(\theta_{t+N-1}) - \mathcal{L}_{main}(\theta_t) - \alpha\nabla_{\theta_{t+N-1}}\mathcal{L}_{main}(\theta_{t+N-1})^T\nabla_{\theta_{t+N-1}}\mathcal{L}(\theta_{t+N-1})$$

$$\vdots$$

$$\approx -\alpha\sum_{j=0}^{N-1}\nabla_{\theta_{t+j}}\mathcal{L}_{main}(\theta_{t+j})^T\nabla_{\theta_{t+j}}\mathcal{L}(\theta_{t+j})$$

Next, we want to update $\boldsymbol{w}$ by calculating $\nabla_{\boldsymbol{w}}\mathcal{V}_t^N(\boldsymbol{w})$, which requires differentiating through the optimization process. To avoid this cumbersome computation in a online process, we ignore all the higher order terms, essentially assuming that a small perturbation of $\boldsymbol{w}$ only affects the immediate next step. With this approximation, we get that $\forall i = 1, ..., K$:

$$\nabla_{w_i}\mathcal{V}_t^N(w_i) \approx -\alpha\sum_{j=0}^{N-1}\nabla_{\theta_{t+j}}\mathcal{L}_{main}(\theta_{t+j})^T\nabla_{\theta_{t+j}}\mathcal{L}_{aux,i}(\theta_{t+j}). \tag{6}$$

We call this approach Online Learning for Auxiliary losses (OL-AUX). The full algorithm is described in Algorithm 1. As an implementation detail, to balance the norm of the gradient between different losses, we adopt the Adaptive Loss Balancing technique [30] and wrap all the auxiliary task losses inside a log operator. Figure 1 provides an illustration of the pipeline for computing all the individual losses that constitute $\mathcal{L}(\theta_t)$.

The benefit of the N-step update, compared to the one-step update, comes from two sources. First, as shown in Eqn. 5, the N-step objective considers the long-term effect on the main task loss of updating the weights $\boldsymbol{w}$, which aligns with our longer term goal. Second, as shown in Eqn. 6, the N-step method also averages the auxiliary weight gradient over more $\theta$ update iterations, which will compute a less noisy gradient. Ablation experiments are shown Section 5 to differentiate between these effects, and we show that both of these effects contribute to our performance.

---

**Algorithm 1** Learning with OL-AUX

---

**Input:**
　　Main task loss: $\mathcal{L}_{main}$
　　K auxiliary task losses: $\mathcal{L}_{aux,1}, \ldots, \mathcal{L}_{aux,K}$
　　Horizon $N$
　　Step size $\alpha, \beta$
**Initialize** $\theta_0, \boldsymbol{w} = \mathbf{1}, t = 0$,
**for** $i = 0$ **to** $TrainingEpoch - 1$ **do**
　　Collect new data using $\theta_t$
　　**for** $j = 0$ **to** $UpdateIteration - 1$ **do**
　　　$t \leftarrow i \cdot UpdateIteration + j$
　　　Sample a mini-batch from dataset
　　　$\mathcal{L}(\theta_t) \leftarrow \log\mathcal{L}_{main}(\theta_t) + \sum_{i=1}^K w_i \log\mathcal{L}_{aux,i}(\theta_t)$
　　　$\theta_{t+1} \leftarrow \theta_t - \alpha\nabla_{\theta_t}\mathcal{L}(\theta_t)$
　　　**if** $t + 1 \bmod N == 0$ **then**
　　　　$\nabla_{w_i}\mathcal{V}_{t-N+1}^N(w_i) \leftarrow -\alpha\sum_{j=0}^{N-1}\nabla_{\theta_{t-j}}\log\mathcal{L}_{main}(\theta_{t-j})^T\nabla_{\theta_{t-j}}\log\mathcal{L}_{aux,i}(\theta_{t-j})$
　　　　(Based on equation 6)
　　　　$\boldsymbol{w} \leftarrow \boldsymbol{w} - \beta\nabla_{\boldsymbol{w}}\mathcal{V}_{t-N+1}^N(\boldsymbol{w})$

---

## 5  Experiments

In the following experiments, we aim to answer the following questions:

- Can OL-AUX adapt the weights online to optimize for the main task and ignore harmful auxiliary tasks?

- How much can we improve sample efficiency by leveraging a set of diverse auxiliary tasks?

- Is dynamically tuning the weights of the auxiliary tasks important to the achieved sample efficiency, compared to using a fixed set of weights?

- Is it beneficial to adapt the auxiliary task weight based on its longer term effect, i.e. N-step update (Section 4.2) compared to the 1-step update (Section 4.1)?

We first answer some of the questions on a simple optimization problem. Then, we empirically evaluate different approaches on three Atari games and three goal-oriented reinforcement learning environments with visual observations, where the issue of sample complexity is exacerbated due to the high dimensional input.

## 5.1 Ignore Harmful Auxiliary Tasks

We first show in a simple optimization problem that OL-AUX is able to ignore an adversarial auxiliary task and find the global optimum of the main task. Later we will also show this in more complex and realistic experiments. In this example, the main task is to find $x, y$ to minimize the loss $L_0(x, y) = x^2 + y^2$. There are two auxiliary losses, $L_1(x, y) = (x - 0.5)^2 + (y - 0.5)^2$ and $L_2(x, y) = -L(x, y)$. Clearly $L_2$ is an adversarial auxiliary task as its gradient always points in the opposite direction to the optima of the main task. On the other hand, $L_1$ is a useful auxiliary task. The baseline we compare to is to optimize the total loss $L = L_0 + w_1 L_1 + w_2 L_2$ with a fixed weight $w_1 = w_2 = 1$, using gradient descent. Our method is OL-AUX as described in Algorithm 1, using $N = 1$ and without applying the log operator in front of the loss as we do not need to balance the gradient for this simple example. The results are shown in Figure 2. The fixed weight baseline converges to a sub-optimal point as the main task loss is canceled out with the adversarial auxiliary task. On the other hand, OL-AUX finds the optimum of the main task from different starting points. From the auxiliary task weights during training of OL-AUX, we can see that the weight of auxiliary task $L_1$ first increases as it helps to learn the main task and later decreases to focus on the main task. On the other hand, the weight of auxiliary task $L_2$ quickly decreases in the beginning.

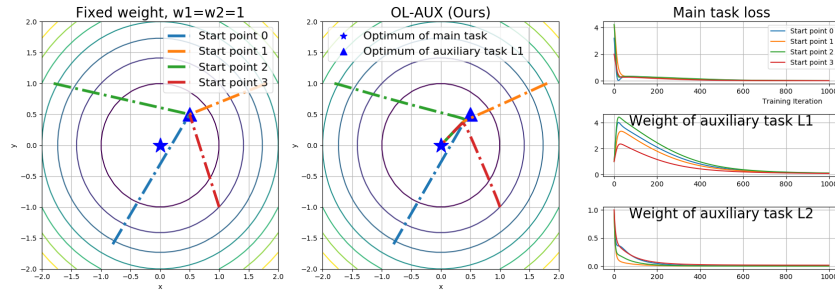

Figure 2: A simple optimization problem to show that our OL-AUX is able to ignore harmful auxiliary tasks. Left and middle show the trajectories of the optimization from four starting points using fixed weights and OL-AUX respectively. Right shows the loss and auxiliary task weights during training for OL-AUX.

## 5.2 Auxiliary Tasks

For more complex visual manipulation, we consider the set of auxiliary tasks shown in Table 1. For a more detailed description of the auxiliary tasks, see Appendix B.

## 5.3 Base Learning Algorithm and Environments

We evaluate our method in two scenarios: 1) Reinforcement learning with A2C [35] as the base learning algorithm, evaluated on three Atari games [36]: Breakout, Pong and SeaQuest. 2) Goal-conditioned reinforcement learning using DDPG [3] with hindsight experience replay [37], evaluated on three visual robotic manipulation tasks simulated in MuJoCo [38]:

| Auxiliary Task | Description |
|---|---|
| Forward Dynamics [12] | Given current visual observation and action, predict latent space representation of next observation. |
| Inverse Dynamics [12] | Given consecutive image observations, predict the action taken. |
| Egomotion [13] | Given raw and transformed visual observation, predict the transformation applied. |
| Autoencoder | Reconstruct visual observation from latent space representation. |
| Optical Flow [14] | Given two consecutive visual observations, predict the optical flow. |

Table 1: Brief description of the auxiliary losses used.

- **Visual Fetch Reach** (OpenAI Gym [39]). The goal is to move the end effector of a Fetch Robot to a randomly sampled 3D location.

- **Visual Hand Reach** (OpenAI Gym [39]). A target hand pose with the positions of the five fingers of a 24 DOF Shadow hand is randomly sampled from 3D space; the policy is required to control the hand to reach the corresponding positions of all five fingers. The policy outputs motors commands for the 24 DOFs of the hand.

- **Visual Finger Turn** (DeepMind Control Suite [5]). A policy needs to control a 3 DOF robot finger to rotate a body on an unactuated hinge. The agent receives a positive reward if the body tip coincides with a randomly sampled target location.

For all the manipulation environments, the goal is an RGBD image with objects in the desired configuration. We use sparse rewards specified by the $L_2$ distance of the underlying ground truth state from the goal state. For hindsight experience replay, with a probability of 0.9 we relabel the original goal with another observation from a future time step of the same episode. More details on the environments and the algorithm used can be found in Appendix A and B.

In both cases, we use Adam as our optimizer. While this creates a discrepancy between our theoretically-derived gradient and the gradient used in practice, we do not find this to be a big issue during our experiments."

## 5.4 Baselines Compared

We compare the following approaches:

1. **No Auxiliary Losses** This is our base learning algorithm without using any auxiliary tasks.
2. **Gradient Balancing** This baseline combines all the auxiliary tasks with the same weight of 1 but uses adaptive loss balancing [30] to balance the norm of the gradient for different auxiliary tasks.
3. **Cosine Similarity** This baseline combines the gradients from the auxiliary tasks and the main task based on their cosine similarities [34]. Specifically, $\nabla_\theta \mathcal{L}_{aux,i}$ is added to $\nabla_\theta \mathcal{L}_{main}$ to update the shared parameter $\theta$ only if $\cos(\nabla_\theta \mathcal{L}_{main}, \nabla_\theta \mathcal{L}_{aux,i}) \geq 0$.
4. **OL-AUX** This is our method as described in Algorithm 1, with N=5 (OL-AUX-5).
5. In Appendix E, we compare with grid search and heuristic fixed weights in the Visual Hand Reach task.

## 5.5 Online Learning of Auxiliary Task Weighting

The learning curves of all methods are shown in Figure 3. All experiments are run with five different random seeds and the shaded region shows the standard deviations across different seeds. We can see that, in all environments, using auxiliary tasks with gradient balancing [30] gives consistent improvement over not using auxiliary tasks. By adapting the auxiliary task weights online, our OL-AUX-5 method shows even more improvement and requires fewer than half of the samples to converge in all the environments.

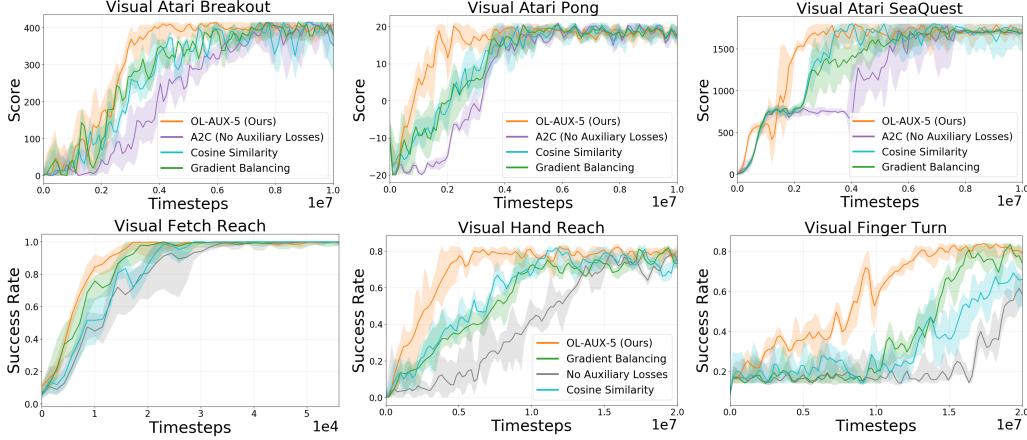

Figure 3: The training curves of our method compared to other baselines. For the manipulation environments, the y-axis shows the percentage of the time that the goal is reached.

For the manipulation environments, we further compare with using only a single auxiliary task along with gradient balancing in Figure 4. We can see that using a single auxiliary task usually gives marginal improvement over the baseline. On the other hand, our method, by leveraging the combination of all auxiliary tasks, always performs better or as well as the best single auxiliary task. Additionally, we can see that the importance of the auxiliary task depends on the RL task. For example, inverse kinematics is the best single auxiliary task for the Visual Hand Reach environment but does not help much in the Visual Finger Turn environment. Our method is able to exploit the best combination of the auxiliary tasks without prior knowledge about the relationships among the auxiliary tasks as well the relationship between the auxiliary tasks and the main RL task.

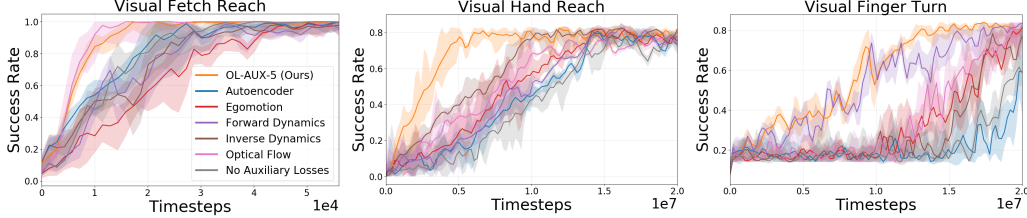

Figure 4: The training curve for our method (which combines multiple auxiliary losses) compared to using each individual auxiliary loss one at a time. The y-axis shows the percentage of the time that the goal is reached.

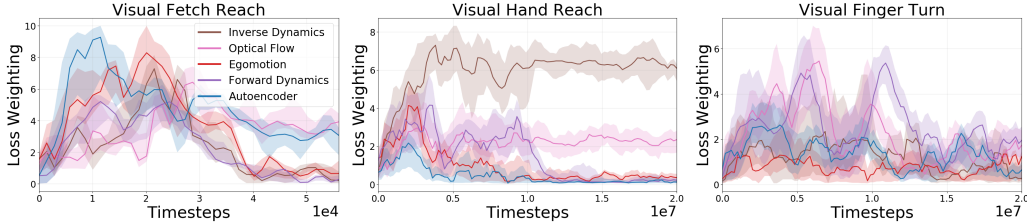

Figure 5: Change of the weights of the auxiliary tasks during training.

In Figure 5, we show how the weights of all the auxiliary tasks change during training for OL-AUX. Looking at the weight of an auxiliary task alongside the single auxiliary task ablation in Figure 4, we can see that they often align. For example, inverse kinematics is the best single auxiliary task in Hand Reach and it also retains a large weight when combined with other auxiliary tasks. There are also exceptions: In Finger Turn, optical flow does not work well as a single auxiliary task. But when combined with other auxiliary tasks, it still has the largest weight for a small amount of time; In Hand Reach, optical flow performs well as a single task but when combined with others, the weight is kept at a relatively low level. This shows that our method is able to take advantage of the auxiliary tasks

that best suits the RL task at hand, while taking into consideration the interplay of different auxiliary tasks, without any prior knowledge.

## 5.6 Auxiliary Task Gradients should Provide Long-term Guidance

Our N-step update method incorporates the idea that the auxiliary tasks should be used to decrease the loss of the main task in the long term. To verify that this long-term reasoning is important, we compare OL-AUX-5 with OL-AUX-1 where the weights are updated every time step (as in Sec. 4.1). For OL-AUX-1, we scale the learning rate $\beta$ down by a factor of 5 to make a fair comparison, as it updates $w$ more frequently. The results are shown in Figure 6. As shown, OL-AUX-1 performs much worse than OL-AUX-5, providing evidence of the importance of using auxiliary tasks to provide long-term guidance for reinforcement learning.

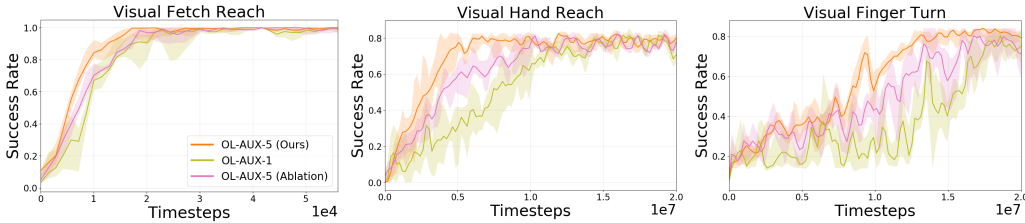

Figure 6: Learning curves comparing different ablation methods.

However, as discussed Sec. 4.2, there are two reasons why our method might outperform the OL-AUX-1 baseline: our method takes into account the long-term effect of the auxiliary weight update on the main objective; also, our method averages the auxiliary weight gradient over more updates on $\theta$, which will result in a less noisy gradient update. To isolate the influence of each of these effects, we perform another ablation experiment. In this ablation, we perform a one-step gradient update using Eqn. 4, but only update $w$ every N (N = 5) steps. When updating $w$, we use N times as much data to compute the gradient, and we use the same learning rate $\beta$ as OL-AUX-5. This makes the algorithm the same as OL-AUX-5 in terms of update frequency, learning rate, and the amount of data used for the update. The plots from this method are labeled **OL-AUX-5 (Ablation)** in Figure 6. The gap between OL-AUX-5 and OL-AUX-5 (Ablation) shows the effect of "long-term reasoning" while the gap between OL-AUX-5 (Ablation) and OL-AUX-1 shows the effect of "gradient smoothing". We can see that both factors contribute to the gained improvement in training time.

## 6 Conclusions

In this work, we have shown that dynamically combining a set of auxiliary tasks can give a significant performance improvement for reinforcement learning from high dimensional input. Our method adaptively adjusts the weights for the auxiliary tasks in an online manner, showing large improvement over a baseline method that treats each auxiliary task as equally important. Our method uses the idea that auxiliary tasks should provide a gradient update direction that, in the long term, helps to decrease the loss of the main task, showing large improvement over one-step reasoning. The task weights we learn with OL-AUX indicates the optimal amount of knowledge to transfer between the auxiliary tasks and the main task. For future works, OL-AUX can also extend to the multi-task learning. Additionally, it would be interesting to explore if the task weights we learn with OL-AUX can transfer across different domains.

**Acknowledgments**

We would like to thank Ben Eysenbach for helpful feedback on the workshop version of the paper and Wen Sun, Lerrel Pinto, Siddharth Ancha, Brian Okorn for useful discussions.

This material is based upon work supported by the United States United States Air Force and DARPA under Contract No. FA8750-18-C-0092, National Science Foundation under Grant No. IIS-1849154 and USDA Specialty Crop Research Initiative Efficient Vineyards Project 2015-51181-24393.

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
