[Supplementary Material]

# A    Environment Details

Details of the MuJoCo manipulation environments are listed below:

- **Visual Fetch Reach** (OpenAI Gym [1]). The goal is to move the end effector of the Fetch Robot to a randomly sampled 3D location. The agent receives a positive reward if the end effector position is within 5 cm from the goal position. The observation for our policy is the current RGBD image and the goal is an RGBD image of the end effector in the goal position.

- **Visual Hand Reach** (OpenAI Gym [1]). A target hand pose with the positions of the five fingers of a 24 DOF Shadow hand are randomly sampled from a 3D space; the policy is required to control the hand to reach the positions of all five fingers. The policy outputs motors commands for the 24 DOFs of the hand. The agent receives a positive reward if the total distance of the end points of the fingers from the goal positions is less than 3 cm. The observation for our policy is the current RGBD image and the goal is an RGBD image of the hand in the desired pose.

- **Visual Finger Turn** (DeepMind Control Suite [2]). A policy needs to control a 3 DOF robot finger to rotate a body on an unactuated hinge. The agent receives a positive reward if the body tip coincides with a randomly sampled target location. The observation is three stacked RGBD frames and goal is an RGBD image with the body rotated in the goal position.

During evaluation, for both environments, a binary sparse reward is given at each time step. A positive reward $R_+ = 1$ is given when the goal is reached, i.e. $||s_{t+1} - s_g|| \leq \epsilon$ and a negative reward $R_- = -1$ is given otherwise. Other environment details are summarized in Table 1. Visual Fetch Reach and Visual Hand Reach environments are taken from OpenAI gym robotics environments. Visual Finger Turn is taken from DeepMind's DM Control Suite.

| Environment | Observation Dimension | Goal Dimension | Action Dimension | Horizon (T) | $\epsilon$ (m) |
|---|---|---|---|---|---|
| Visual Fetch Reach | 100x100x4 | 3 | 3 | 50 | 0.05 |
| Visual Hand Reach | 100x100x4 | 15 | 24 | 50 | 0.03 |
| Visual Finger Turn | 100x100x12 | 2 | 2 | 75 | 0.07 |

Table 1: Environment details including observation dimensions, goal dimensions in state space (though we use image goals with the same dimension as the observation for learning), the time horizon, and the distance threshold for defining task success.

# B    Auxiliary tasks

In the context of manipulation from visual observations, we consider a set of auxiliary tasks described below:

1. **Forward Dynamics** [3]: This task enforces forward consistency in the learned latent representation. Given the current visual observation and the action, the network is asked to predict the latent representation of the next state. The loss is defined as the following:

$$\mathcal{L}_{fk} = ||f_{fk}(e(o_t; \phi), a_t; \phi_{fk}) - e(o_{t+1}; \phi)||_2^2,$$

where $f_{fk}$ and $e$ are the latent space forward model and the CNN encoder respectively, $o_t$ is an observation, and $a_t$ is an action.

2. **Inverse Dynamics** [3]: Given two consecutive image observations, this task predicts the action taken. The loss is specified by
$$\mathcal{L}_{ik} = ||f_{ik}(e(o_t; \phi), e(o_{t+1}; \phi); \phi_{ik}) - a_t||_2^2.$$
where $f_{ik}$ is the latent space inverse dynamics model.

3. **Egomotion** [4]: Given an image observation and a random transformation of this image, the network needs to predict the performed transformation. This task forces the network to learn visual correspondences between the transformed image and the original image. In our experiments, the transformation is constrained to be a planar rotation with a degree of $\theta \in [-30°, 30°]$. The transformed image is then clipped and scaled to have the same size as the original image. The input to the egomotion prediction network also shares the convolutional features with other tasks, including the main task. The loss is defined as:

$$\mathcal{L}_{eg} = ||f_{eg}(e(o_t; \phi), e(\mathcal{T}o_t; \phi); \phi_{eg}) - \theta||_2^2,$$

where $\mathcal{T}$ is the transformation.

4. **Autoencoder**: This task aims to reconstruct the image observation given the latent representation $e(o; \phi)$. It enforces a representation that preserves the information in the original observation as much as possible. The loss is defined as:

$$\mathcal{L}_{ae} = ||f_{ae}(e(o_t; \phi)); \phi_{ae}) - o_t||_2^2$$

5. **Optical Flow** [5]: Between every two consecutive visual observations, we first compute the visual representation of optical flow using Farneback's [6] algorithm. Then, the network needs to predict the optical flow result from the latent representation of the two images. This task encourages a latent representation that focuses more on the moving pixels and could be helpful for the object manipulation tasks. The optical flow loss is defined as:

$$\mathcal{L}_{op} = ||f_{op}(e(o_t; \phi), e(o_{t+1}; \phi)); \phi_{op}) - \mathrm{FK}(o_t, o_{t+1})||_2^2,$$

where $f_{op}$ denotes the optical flow prediction network to be learned and FK denotes the Farneback's algorithm. Target optical flow representation for two NxNxD frames is an NxNx1 mask where the value of each pixel in the mask represents the strength of the flow vector at that location.

## C  Hyper-parameters

All the experiments are run for five random seeds. The hyper-parameters for the manipulation environments are described in Table 2. For all experiments, the parameters of the encoding convolution layers are shared among the observation input and the goal input along with all visual information for the auxiliary rewards.

| Parameter | Value |
|---|---|
| positive reward ($R_+$) | 1 |
| negative reward ($R_-$) | -1 |
| optimizer | Adam [7] |
| learning rate | 0.001 |
| learning rate for $\boldsymbol{w}$ ($\alpha\beta$) | $0.005 \times N$ (where N comes from OL-AUX-N) |
| discount ($\gamma$) | $\frac{T}{T-1}$ |
| target network smoothing ($\tau$) | 0.98 |
| nonlinearity | tanh |
| replay buffer size | $5 \cdot 10^3$ |
| minibatch size | 64 |
| network architecture | 4 convolution layers with 64 filters each followed by 3 hidden layers with 256 neurons for each |

Table 2: Summarized hyper-parameters. We did not tune these hyper-parameters.

# D  Training Details

To generate success rate plots, after 400 training cycles, we evaluate the policy for 25 episodes. Each plot is generated from running the same experiment for two random seeds. The solid line represents the median across different random seeds while the shaded region represents the 25 and 75 percentile. Auxiliary reward weights ($\boldsymbol{w}$) are recorded at the same rate as the evaluation cycles. Each experiment was run on a compute node with an Intel Xeon processor and NVIDIA GeForce Titan X Pascal 12 GB GPU. We used 8 parallel threads to generate rollouts for each cycle.

# E  OL-AUX Outperforms Hand-tuned Weights

As doing grid search over all auxiliary tasks weights are infeasible due to the large amount of computation, we try to approximate the optimal hand-tuned weights in two ways: 1) In the single auxiliary task case, we use grid search to find the optimal weight for that auxiliary task. 2) In the multiple auxiliary tasks case, we use the final weights learned by OL-AUX-5 as the optimal hand-tuned weights. We compare with these two baselines in the Visual Hand Reach environments. The results are shown in Supplementary Figure 1. We can see that in either cases, OL-AUX performs the optimal hand-tuned weights. This shows that OL-AUX is able to eliminate the need to hand-tune the auxiliary task weights. Additionally, the grid search experiment shows that it is beneficial to have dynamic weights instead of static weights.

(a) Grid search is done by discretizing the weight space into 10 values, training an agent for each of the value it and pick the best one.

(b) Hand tuned weights are obtained by first training OL-AUX-5 to convergence and take the final weights.

Supplementary Figure 1: Comparison of our method to grid search or hand tuned fixed weights.