[Reviews · NeurIPS 2019]

Reviewer 1



originality: training reinforcement learning with auxiliary losses can trace back to "reinforcement learning with unsupervised auxiliary tasks" however difficulties in cooperating auxiliary losses with main objectives block this idea from being widely applied. Unlike previous works where influences from auxiliary losses depend on manually set hyper parameters or cosine similarity on gradients, this paper originally propose to differentiate main objective using Taylor approximation involving gradients of auxiliary losses. quality: the paper show detailed derivation of proposed update rules. Even though the remaining derivation still goes, I am not sure if the application of the finite difference approximation in the first line of equation 3 is appropriate as the difference depends on the step size alpha, which can be relatively large. The paper also provide n-step version of suggested weight update. Given assumption listed in the paper, this extension should be reliable. clarity: the writing of this paper is basically clear. My only concern is about the adaptive loss balancing technique. More explanation on this technique and the motivation of using it on this work can be helpful. Is it widely used in similar setting or does it have any limit? Does the advantage of proposed method depend on this technique or any other auxiliary losses method ,like cosine similarity, can also benefit from it? significance: in two of three listed experiments, proposed method clearly outperforms others. In the remaining one, the proposed method is still better than others even though the difference is not as big as other two experiments. Authors also provides experiments supporting the advantage of their n-step update over one step update. Again the difference in the experiment of "Visual Fetch Reach" is not as big as the other two.

Reviewer 2



# Update after the rebuttal I appreciate the authors addressing most of my concern regarding 1) hand-tuned fixed baseline, 2) more results on Atari, and 3) handling harmful auxiliary tasks. I think the results are much more comprehensive now. I raised my score accordingly. # Originality - This paper proposes a novel and interesting method to adapt the weights over different auxiliary objectives. If I understand the main idea correctly, the proposed method can be interpreted as a gradient-based meta-learning method (e.g., MAML) in that the algorithm finds the gradient of the main objective by taking into account the parameter update procedure. It would be good to provide this perspective and also review the relevant work on meta-gradients for RL (e.g., MAML [Finn et al.], Meta-gradient RL [Xu et al.], Learning intrinsic reward [Zheng et al.]). Nevertheless, I think this is a novel application of meta-gradient for tuning auxiliary task weights. # Quality - The proposed method based on gradient similarity between main loss (RL) and auxiliary loss sounds sensible. - The results on the domain considered in this paper look promising (outperforming baselines). However, the overall results are not comprehensive enough to show the benefit of the proposed approach for the following reasons. 1) This paper does not have an important baseline: hand-tuned but fixed auxiliary weights. This paper claims that the optimal auxiliary weights may be different at different learning stages, but this claim is not empirically shown in the paper because of the lack of such a baseline. 2) The domain and the auxiliary tasks considered in this paper are different from the ones used in the baselines (gradient balancing, cosine similarity). So, it is hard to see how significant the improvement is. It would be much more convincing to show results on the same setup as in the previous papers. 3) It would be much more convincing to show broader experimental setups. For example, all the auxiliary tasks in the experiment happen to have a positive effect in the experiments (as the learned weights are all positive). It would be interesting to see if the proposed method can also deal with the negative effect of auxiliary tasks. Also, the proposed method can be used in the multi-task learning scenario, where other tasks are considered as auxiliary tasks, It would be again interesting to automatically adjust the weights over different tasks to prevent negative transfer while encouraging positive transfers. Finally, it is not clear if the proposed method is specific to RL. If yes, it would be good to have a short discussion on why this is specifically useful for RL. If no, it would be good to show some empirical results on supervised learning as well as RL. Showing this would make the contribution of the paper clear. 4) (relatively minor) The main algorithm used in this paper (HER) is specifically designed for goal-based RL, where a goal is given as an additional input. It would be better to consider a more general RL problem/algorithm instead of goal-based RL. # Clarity - This paper is well-written and easy to follow. - It would be good to discuss a connection to meta-learning in the method section and the related work. # Significance - The idea behind this paper is novel and interesting. However, the experimental settings are quite limited to one specific domain (visual MuJoCo) and one specific method (hindsight experience replay) with a specific set of (positive) auxiliary tasks. Thus, it is not clear if this paper is significant enough to get interests from the Deep RL community.

Reviewer 3



[summary] This paper proposes a method for learning auxiliary task weighting for reinforcement learning. Auxiliary tasks, if chosen appropriately, can accelerate the acquisition of useful representations, leading to sample-efficient learning. Discovering the right auxiliary tasks is an important research question and weighting auxiliary tasks with respect to their utility is an important part of the answer. This paper argues that an auxiliary task has high utility if it helps decrease the loss of the main task. On the basis of this principle, a gradient-based rule to update the weighting online is derived and empirically evaluated on simulated manipulation tasks. [decision] I’m leaning towards rejecting this paper. This paper is addressing an important problem, the proposed solution is interesting and could indeed end up being a useful contribution to the literature. Nevertheless, I am not yet convinced if the proposed approach is an effective solution especially if the learning signal for the main task is sparse. The empirical section does not alleviate my concerns especially because the paper does not provide any justification for the chosen hyper-parameters and the learning curves are averaged over only two independent runs. Given the high-variance of deep reinforcement learning methods, I cannot measure the contribution of the new method with the limited empirical analysis provided in the paper. [explanation] I agree with the general principle that an auxiliary task is useful if it helps decrease the loss of the main task, and I like how this idea is formalized and used to derive a gradient-based update for task weightings. The resulting update also makes intuitive sense: the weighting should be updated so as to maximize the cosine similarity between the gradients of the auxiliary tasks and that of the main task. If the reward signal is dense, this should be able to weed out auxiliary tasks which negatively interfere with the main task. However, I’m concerned if this update would be able to find the appropriate weightings if the reward for the main task is encountered very sparsely – a setting where auxiliary tasks have been found to be most useful. If the reward signal is sparse, it would be significantly harder to assess the effect of auxiliary tasks on the loss of the main task. This paper mischaracterizes the environments it uses when it says that the environments have sparse rewards. A reward of -1 at every step until the agent finds the goal is dense and provides a rich training signal. The proposed method would be a lot more compelling if this paper includes experiments on sparse reward tasks e.g. a positive reward when the agent reaches the goal and zero elsewhere. The empirical work done in this paper indicates that the method can work. The baselines are appropriate and comprehensive. While the learned tasks weightings are fidgety, in some cases they still manage to separate auxiliary tasks which interfere with the main task from the ones which are actually helpful (Figure 4). This indicates that the method is promising. Nevertheless, I have a few concerns about the experiments. The hyper-parameter setting is presented in the supplementary section but I cannot find any details about how it was chosen. Were these hyper-parameters appropriately tuned for Hindsight Experience Replay? In addition, the experiments were run for only two random seeds. In the last section of supplementary materials, it says “..the solid line represents the median across different random seeds while the shaded region represents the 25 and 75 percentile” (supplementary materials). With just two independent runs, I find this description and the plots in the main paper with shaded lines very misleading. The notion of median and percentiles becomes meaningless if we have only two samples. Given high-variance of deep reinforcement learning methods, this empirical setup casts doubts on the conclusion drawn in this paper. The proposed method should be contrasted with Du et al. gradient similarity method [1]. While there is a passing comment about that method in this paper and it is also included as a baseline, there are indeed significant differences and this paper would benefit from having a brief discussion around those differences. [Small things] Equation 3 does not hold if we use an adaptive step size optimizer such as Adam (which is what is used in the experiments). This is fine but it should be mentioned in the paper. In the paragraph just before the conclusion, it is stated that OL-AUX-5 shows the effect of “long-term reasoning”. I think this an overreach as only 5 steps shouldn’t be sufficient to tell apart good auxiliary tasks from the bad ones, especially when the optimization landscape is complicated and the problem is non-stationary (which is usually the case in reinforcement learning). In any case, this paper should avoid suggesting that any sort of “reasoning” is being done here. The paper uses HER as the baseline. It is indeed a strong baseline, but the algorithm is relatively non-standard and complex. I suggest something like DDPG should be used as the default baseline. In figure 1, there is a box for “visual auxiliary info” but it is not referred anywhere else in the paper. I’m not sure what it represents and it feels extraneous. If this doesn’t serve a clear purpose, I suggest that it should be removed from the figure. [1] Yunshu Du, Wojciech M Czarnecki, Siddhant M Jayakumar, Razvan Pascanu, and Balaji Lakshminarayanan. Adapting auxiliary losses using gradient similarity. arXiv preprint arXiv:1812.02224, 2018.

[Author Response · NeurIPS 2019]

We thank the reviewers for the positive feedback on our motivation and algorithm. As most of the concerns are on our
empirical study, we now address these concerns with additional experiments and some clarifications:

**More random seeds:** We originally used 2 random seeds as goal conditioned reinforcement learning has a relative low variance from our
experience. We have re-run all the experiments with 5 random seeds and all our results still hold. The updated figure of the submitted paper's Figure 2 is shown on the right.

**Comparison with more baselines:** 1) **Hand tuned, fixed weights:** We compare OL-AUX with hand-tuned weights
either on a single auxiliary task, where the best fixed weight is found with grid search (Figure 1), or on all the auxiliary
tasks where the best fixed weights are the final weights learned by OL-AUX (Figure 2). It shows that our method can
adaptively combine auxiliary tasks and outperforms the best fixed weight. 2) **No gradient balancing:** In our original
experiments, we compare to the cosine similarity method [Yunshu et al. 2018] with gradient balancing added for fair
comparison. We show in Figure 3 that cosine similarity performs worse when gradient balancing is removed.

Figure 1: Comparing handtuned weight, single auxiliary task.

Figure 2: Comparing handtuned weights, all auxiliary tasks.

Figure 3: Effect of gradient balancing on cosine similarity.

Figure 4: Atari seaquest

**Empirical results on benchmark RL tasks:** In Figure
4,5,6, we show that in three benchmark RL environments
in Atari, OL-AUX also outperforms all the baselines. The
base algorithm we use is A2C [Mnih et al. 2016]. All
hyper-parameters are the same as the ones used in the paper or are default to A2C. The same set of auxiliary tasks are also used. This shows that OL-AUX gives significant improvement across different domains.

Figure 5: Atari breakout

Figure 6: Atari pong

**Ability to separate harmful auxiliary tasks:** In Figure 3 of the orignal paper, we show that AutoEncoder is a
harmful auxiliary task for Finger Turn environment. Here, a toy example in Figure 7 with one positive auxiliary
task and one harmful auxiliary task shows that our algorithm is able to avoid adversarial auxiliary tasks without
any prior knowledge. In this 2d example, the main task loss is $L(x, y) = x^2 + y^2$. There are two auxiliary tasks,
$L_1(x, y) = (x - 0.5)^2 + (y - 0.5)^2$ and $L_2(x, y) = -L(x, y)$. Using a fixed weight for auxiliary tasks (Left), the
agent converges to a sub-optimal point. Our method finds the optimum of the main task from different starting points
(Middle). The auxiliary task weights during training for our method (Right) shows that our method is able to ignore $L_2$,
which is a harmful auxiliary task.

Figure 7: Ignoring adversarial auxiliary tasks

Figure 8: Learning with a binary reward of {0, 1} in goal-conditioned RL.

Response to **Reviewer 3**: The MuJoCo environments we tested effectively all have sparse rewards, since scaling and
translating the rewards from {-1, 1} to {0, 1} does not change the ordering of the value function or the optimal policy.
Nevertheless, in Figure 8 we show one experiment using a negative reward of 0 where our method performs just as well.
Additionally, most of our hyper-parameters in the paper are taken from the defaults of Hindsight Experience Replay.

[Meta-Review · NeurIPS 2019]

Reviewers agreed that the paper addresses an important problem in current deep RL research and appreciated the effort put into the rebuttal by the authors. New experiments using the 0/1 reward formulation and a comparison to fixed hand-tuned hyper parameters addressed two of the main concerns raised by reviewers. In the end all three reviewers recommended accepting the paper.